# Spatial Fiscal Interactions in Colombian Municipalities: Evidence from Oil Price Shocks

**Raju Mainali** [1,2]

1 Information Technology and Cybersecurity, College of Business, Missouri State University, Springfield, MO 65897, USA; rajumainali@missouristate.edu
2 Department of Information Technology and Cybersecurity, Missouri State University, 901 S. National Ave., Springfield, MO 65897, USA

**Abstract:** This study provides an empirical investigation of fiscal interactions in the context of a developing country. I examine three fiscal components—budget balance, tax revenue, and public spending—to measure spatial interactions between Colombian municipalities from 2000 to 2010. I am using variables on municipalities' general characteristics, fiscal variables, and variables related to the conflict. I use a quasi-experimental identification strategy exploiting exogenous variation from global oil price shocks that affect Colombian municipalities to different degrees depending on local oil endowments. I find significant spatial interaction in taxes but no significant interaction concerning budget balance and total public spending. This suggests that even though there is local tax competition, municipalities do not mimic their neighbors to decide whether to offset tax changes by changes in borrowing or spending.

**Keywords:** fiscal interaction; budget balance competition; tax competition; expenditure competition; spatial interaction quasi-experimental





## 1. Introduction

Fiscal decentralization reform in developing countries has been advancing, particularly in the last two decades. Developed countries are more decentralized than developing countries, and many countries in the developing world have also decentralized their government structures partly due to the involvement of international organizations, other aid agencies, and government and policy experts. The goal of fiscal decentralization in developing countries focuses primarily on revenue transfer and reassigning expenditure decisions to the sub-national level. Many developing countries have been involved in fiscal decentralization by providing local governments with different fiscal autonomy levels with the possibility of fiscal interaction among local jurisdictions in those countries. The fiscal competition between local governments may arise due to the fiscal autonomy at the local level. The fiscal autonomy includes debt management, tax administration, and budget execution at the local level. Local government budgets are required to be balanced by law in many developing countries. However, in some developing countries, there is no budget balance requirement, resulting in a severe fiscal gap to maintain the budget balance with the controlled revenue capacity of local government. Most developing countries' local governments run a deficit budget instead of a surplus because of their limited revenue source and the degree of revenue autonomy.

Fiscal competition is likely to happen not only between countries but also among local governments. There is evidence of fiscal competition and yardstick competition between Indonesian districts after its decentralization reform (Arze del Granado et al. 2008). The fiscal decentralization reform in China stimulated the local governments' enthusiasm for

developing the region's economy and escalating the fiscal competition between local governments. The fiscal spatial interaction among the local government might help to justify the goal of decentralization in many developing countries. Most of the economic literature on fiscal interaction focuses on four different theoretical mechanisms: tax competition, political yardstick competition/benchmark, bailout competition, and expenditure competition. These spatial fiscal interaction competitions between jurisdictions have both significant positive and negative effects after decentralization.

The first theory describes competition among jurisdictions for a mobile tax base suggested by the Tiebout-type model (Tiebout 1956). In this model, if the tax rates are high compared to those in a neighboring jurisdiction, then firms tend to move away. Zodrow and Mieszkowski (1986) suggest that sub-national governments competing for mobile capital will underinvest in public services; (Buettner 2001) observed the significant competition among local jurisdictions for business tax revenues in Germany. Feld and Reulier (2009) show that corporate and income tax rates affect corporate location choice in Switzerland, suggesting that business competition may ignite a "fiscal war", where jurisdictions reduce their tax rates to attract new business. In Brazilian states, Value Added tax (VAT) were reduced to attract new investment but it did not work properly in practice (Ferreira et al. 2005).

A second explanation for fiscal interaction is the political yardstick competition, which encourages local politicians to mimic their neighbor's policies Shleifer (1985); Besley and Case (1995) found empirical evidence of tax mimicking behaviors across states in the United States; Ladd (1992) found similar behavior among counties. US states' tax rates are influenced by those in neighboring states only where the governor could be re-elected. Bordignon et al. (2003) observed behavior for municipalities in Italy when mayors were running for re-election. There is some empirical evidence of tax mimicking in European countries; Heyndels and Vuchelen (1998) found tax mimicking behavior among municipalities in Belgium, while Allers and Elhorst (2005) did so for municipalities in the Netherlands. Electoral outcomes might be affected by their own tax rate and neighbor's tax rate; Dutch voters seem to penalize incumbents for an anticipated tax rate differential but not for an unanticipated tax rate differential. With these models, a jurisdictional policy change on public expenditure and tax revenue produces a strategic incentive for neighboring jurisdictions to change their own policies. Especially in the context of developing countries, voters are highly curious about their neighboring political leaders and their policies. According to Yu et al. (2016) local Chinese leaders engage in tournament competition. Voters in one jurisdiction use information from other jurisdictions to judge the performance of their own politicians. Colombia is also fairly decentralized and municipality and department (province) executive bodies are democratically elected, which means those elected bodies are more accountable to people[1]. Yardstick competition might be relevant for Colombian municipalities because information problems are likely to be comparable between the jurisdictions.

A third explanation suggests the existence of horizontal interaction between jurisdictions to compete for a bailout (Baskaran 2012). In Colombia, the 1991 constitution gave fiscal autonomy to the subnational government for borrowing and bond issuance. Many municipalities have experienced debt problems in Colombia and expected the central government to help them through bailouts. The chance of bailouts had encouraged local governments to spend and incur debt from financial institutions. There can be competition between municipalities regarding who will get a bailout from the central government first. In Colombia, municipalities were allowed to incur debt, which led municipalities to experience debt problems in the 1990s. The fiscal reforms in 1997 and 2000 strongly addressed borrowing and bailout restrictions for local governments, which helped to reduce the municipalities' debt (del Villar et al. 2013). So, bailout competition between the jurisdiction is not applicable for Colombian municipalities because of the time frame used in this study; however, this could be applicable for other developing countries.

A fourth explanation for expenditure interaction is the expenditure spillover effect. This type of competition exists when a decision made by one government affects the

performance of another government (Gordon 1983) There is a lot of empirical evidence on the expenditure spillover effect, mainly from the United States and European countries. An empirical study showed that a state government's per capita expenditure level was positively affected by the expenditure levels of its neighbors (Case et al. 1993), with evidence that a state's expenditure in the United States on roads, education, and welfare is positively affected by its neighbors' expenditure levels. Kelejian and Robinson (1993) showed that police expenditure was higher when police expenditure in neighboring counties was higher. Cohen and Paul (2003) found that airport expansions in one state have a considerable impact on other states. Empirical evidence shown by Murdoch et al. (1993) concluded that non-residents also enjoy the benefits from recreation expenditure by local governments in California. There is limited research on this issue for developing countries. Ferreira et al. (2005) found an under-provision of public health services in the Rio de Janeiro metropolitan area in Brazil because of failure of the internalized expenditure benefits generated by the city of Rio to spillover to its surrounding cities.

This study's primary purpose is to investigate the spatial interdependence of fiscal variables (budget balance, tax revenue, and public expenditure) on the fiscal outcomes of local jurisdictions. Most of the literature focuses on expenditure competition or tax competition (Fossen and Steiner 2016), but not on budget balance competition. Although there is a large amount of literature on local fiscal competition in North America and Western Europe, little is known about the extent and significance of these fiscal interactions among local governments in developing countries. Thus, this paper aims to fill this gap in the literature by investigating whether jurisdictional competition has been present in Colombia.

The simultaneous influence of neighboring municipalities creates an endogeneity problem in spatial econometric estimation. Most of the literature does not use quasi-experimental identification strategies[2] to deal with this endogeneity. Colombian municipalities receive royalties, which depend on revenue from local oil extraction by a private company. In the case of Colombia, I could use exogenous variation due to global oil price shocks. I utilized the fact that there is variation in the global market price of oil that is exogenous from Colombian municipalities' perspectives because Colombia is a small exporter of oil. Colombian municipalities are exposed to these shocks to different degrees because oil resources[3] are heterogeneously distributed across municipalities. I investigated the fiscal variables' spatial spillover effects using this instrumental variable (IV) approach by controlling for municipality fixed effects and time fixed effects.

I used panel data on 1093 municipalities from 2000 through 2010[4]. The Center for Studies on Economic Development (CEDE) at the University of Los Andes consolidated municipal-level data into a single database. I used data from this data set, including variables on municipalities' general characteristics, fiscal variables, and variables related to conflict and violence. The data is of extraordinary quality in a developing country context[5].

The results show that the spatial interactions of budget balance and total public expenditure are not significantly different from zero when identifying the spillover effects based on the quasi-experimental variation. However, there is evidence of a positive spatial interaction effect between the municipalities for the tax revenue. The results suggest that neighboring municipalities are engaging in tax competition but do not mimic their neighbors to use public spending or borrowing to offset tax changes. Thus, although there is tax competition, no race to the bottom occurs regarding budget balances or local public expenditure.

The rest of this paper is arranged in the following way: Section 2 briefly explains the decentralization process and fiscal trends in Colombia, Section 3 presents the data and methodology, and Section 4 presents the empirical model and strategy. The main results based on the quasi-experimental instrument and quasi-maximum likelihood approaches are discussed in Section 5. In Section 6, I discuss the findings and conclude the analysis.

## 2. Decentralization and Fiscal Trends in Colombian Municipalities

### 2.1. Decentralization in Colombia

Colombian politicians adopted a series of decentralization measures in the 1980s and 1990s that were designed to create meaningful new access points in the political system. Decentralization has shifted the decision-making authority downward from the national government to subnational governments, which has often benefited those who continued to sponsor the conflicts occurring in the country. Throughout Colombia, decentralization has enabled illicit groups to exchange goods and services for political support under the threat of violence. Decentralization worsened rather than improved Colombia's security system because the central government failed to provide security, which was also not decentralized. Lack of policy and accountability at the local level was also one reason for increased armed activity at the local level. Decentralization in Colombia had a negative impact on security.

In Colombia, municipalities, not the departments (or provinces), have been the main actors in recent decentralization. Table 1 shows a brief description of major reform efforts since the 1980s. In the early period of reform, mayors were elected for the first time by a popular vote instead of being appointed by the department governor. In 1991, some changes were made in the constitution, following extended discussion by a national constitution assembly. They defined the functions of different levels of government and established a completely new transfer system. During that period, there was excessive borrowing by subnational governments. To control the subnational borrowing, a "Traffic Light" system[6] was introduced which provided the rating system for subnational governments. The new law was applied in 10 departments and 60 municipalities.

The main aim of this law was to impose tighter controls on resource allocations. These efforts were not considered to be adequate, and a new law[7] was introduced. The Law of 2001 established department and municipality "categories" based on the proportion of their current revenue they could spend freely. Then, it set limits on the amount that departments and municipalities in each category could pay to elected and certain appointed officials (Bird 2012). The law modified the constitutional intergovernmental transfer system. It established a new General System of participation (SGP) and set the transfer level to increase by 2% in real terms annually until 2005, then by 2.5% annually until 2008. The Law of 2001 imposed new limits on all subnational governments' expenditure and established a new adjustment package intended to facilitate debt servicing and repayment. Finally, Law 819 of 2003 introduced a new medium-term budgetary framework system for subnational governments, coupled with new fiscal management and transparency rules, including mandatory credit risk analysis before borrowing. The changes after 2001 brought significant improvements to subnational fiscal balances. There is also evidence that municipalities became more prominent revenue generators than departments.

**Table 1.** Major Decentralization Reforms in Colombia.

| Period | Rational | Explanation |
|---|---|---|
| 1983 | New Federalism | Mayors of municipalities were elected for the first time |
| 1991 | Extended discussion by National Constitution Assembly | The National Constitutional Assembly defined the functions of the different levels of government and established a new transfer system. There are three important guides to decentralization. <br><br> 1. **Article 356:** The central government retained its prominence regarding taxation, but transferred a significant share of this power to the regional governments. <br> 2. A need was introduced to make social expenditure (on health and education) more efficient <br> 3. The local governments may not "spend beyond their means." |
| 1997 | Macroeconomic management at all levels of government "Traffic Light" | To control subnational borrowing, the "Traffic Light" **Law 358 of 1997** was introduced, which provides a rating system based on the ratio of interest to budgetary current account saving and debt to the current revenue. |

| Period | Rational | Explanation |
|--------|----------|-------------|
| 2001 | Law 617 of 2001 | • This is the main intergovernmental transfer reform. The constitutional amendment explicitly imposed new limits on administrative expenditure on all local governments and established a new adjustment package intended to help with debt servicing and repayments. This law of 2001 established "Categories" of department and municipalities based on the proportion of their current revenue they could spend freely, and then set limits on the amount that department and municipalities in each category could pay to elected and certain appointed officials based on their categories. Following are the classified categories.<br>• **Special Category**: Populations over 500,000 and incomes over 400,000 LMWs (50%)<br>• **Category 1**: A population of 100,001 to 500,000 and income of 100,001 to 400,000 LMWs (65%)<br>• **Category 2**: A population of 50,001 to 100,000 and income of 50,001 to 100,000 LMWs (70%)<br>• **Category 3**: A population of 30,001 to 50,000 and income of 30,001 to 50,000 LMWs (70%)<br>• **Category 4**: A population of 20,001 to 30,000 and income of 25,001 to 30,000 LMWs (80%)<br>• **Category 5**: A population of 10,001 to 20,000 and income of 15,001 to 25,000 LMWs (80%)<br>• **Category 6**: A population of 10,000 or less and income of 15,000 or less LMWs (80%)<br><br>In 2001, a new **General System of Participation (SGP)** was established, which caused the level of transfer to be increased by 2% in real terms annually until 2005 and then by 2.5% annually until 2008. |
| 2003 | Law 819 of 2003 | Law 819 of 2003 introduced a new medium-term budgetary framework system for subnational governments with new fiscal management and transparency rules, including mandatory credit risk analysis before borrowing. Law 819 requires both the central administration and local government to present a 10-yearmacroeconomic framework each year. |

### 2.2. Colombian Municipalities' Revenue Sources

Municipalities in Colombia generate their revenue through three different sources. The primary revenue source is transferred from the central government, which accounts for 63% of municipalities' total revenue. Tax is another revenue source that contributes to an average of 44% of current receipts and 13% of total revenue. In Columbia, different levels of government collect different taxes. The federal government collects income tax, value-added tax (VAT), and tax on international trade while departments collect taxes on liquor and beer, and municipalities collect business tax (17%), property tax (34%), and a petrol surcharge (22%). Property tax is one of the most significant taxes at the municipality level, levied on all real estate's cadastral value in the municipalities. The third source of revenue for municipalities is the royalties received from natural resources such as oil, coal, and others. The most important source that highly contributed to royalties between 2005 to 2011 was extraction of oil, which contributed towards 69% of total royalties, followed by coal with 23% (Martinez 2019). Firms pay these royalties to the central government according to a set of fixed resource-specific formulas of the form:

$$\text{Royalty} = \text{Output} * (\text{USD}) * \text{Exchange rate} (\text{COP}/\text{USD}) * \text{Royalty rate}. \qquad (1)$$

The primary goal of royalties is to distribute the resources generated from the exploitation of natural resources. After the amount is determined using the above formula, the royalties are distributed to departments and municipalities involved in production

and transportation. Currently, 28 departments and 328 municipalities receive royalties (Inter-American Development Bank). Also, the municipalities' expenditure of natural resource royalties is heavily regulated. By law, at least 75% of royalties must be spent on public services like education, health, drinking water, and sanitation until targets are meet.

*2.3. Sub-National Spending and Debt*

Colombia has become the most decentralized Latin American country among non-federal types, such as Argentina and Brazil. The decentralization process started in the early 1960s in Colombia, and the 1991 constitution vastly accelerated the process. The centralized tax collection method and decentralized spending decision created a risk of fiscal imbalance because the government's level did not fully internalize the cost of its spending. The federal government had set a formula for the spending capacity of local governments. About 80% of budgetary allocation to department and municipalities had to be spent on health and education. Due to unclear spending responsibilities, different government levels created fiscal imbalance problems at the local government level.

The 1991 constitution provided increased autonomy to sub-national governments over borrowing and bond issuance. They could borrow for current spending as long as the ratio of debt servicing to the current income was below 30%, and municipalities could incur debt. Also, Article 364 of the constitution stated that sub-national government debt should not exceed its payment capacity. After the 1991 constitution gave autonomy to the sub-national governments, they lacked fiscal discipline and faced aggressive behavior from banks. This produced a rapid increase in sub-national government indebtedness, especially in 1992–1994, the term of the first elected governors. On top of the growing transfer, sub-national governments ran current fiscal deficits, and new municipalities were created to gain access to transfers. During 1994–1999, the number of municipalities increased from 745. In 1997, Law 358 introduced the constitutional mandate of limiting debt to payment capacity by introducing the "Traffic Light" concept based on the indicator of liquidity (interest payment/operational saving) and solvency (debt/current revenue). They labeled the three-traffic light symbols as the green light zone, yellow light zone, and red-light zone with indicators (liquidity and solvency)[8]. Municipalities that were in the "green light zone" had total autonomy to contract new credit. A performance agreement needed to be signed when the municipalities were in a "red light zone" or, in some cases, in a yellow light zone. The performance agreement was based on negotiations between municipalities and financial institutions and was reviewed and monitored by the Division of Fiscal Support (DAF) at the Ministry of Finance. In this agreement, municipalities had to meet some series of targets within a predetermined time frame. The objectives consisted of an increase in own revenues, control of expenditure generation of current savings, and an improved debt profile.

**3. Data**

Colombia is composed of 32 regions (or departments) with 1122 municipalities. Each department and municipality are under the supervision of a governor or mayor, respectively. I used panel data on 1093 municipalities for the 2000–2010 period. The Center for Studies on Economic Development (CEDE) at the University of the Andes consolidated municipal level data into a single database. Data includes variables on general characteristics of municipalities, fiscal variables, oil endowments, time-series data on international oil price, geo-information (to create a spatial weight matrix), and conflict data.

In regression analysis, municipal-level panel data for the period 2000–2010 was used. These data have been already used to analyze spending interaction (Fossen et al. 2017), the effect of fiscal decentralization (Soto et al. 2012), income shocks (Dube and Vargas 2013), political stability (Acemoglu et al. 2013b), and revenue and government performance (Martinez 2019). The variables are expressed in real per capita Colombian pesos (COP) with the base year as 2008. The dataset has high-quality local-level data with more informative variables. The dependent variables for this empirical analysis are budget balance, total

public expenditure, and tax revenue per capita. The control variables are population, population squared, the fiscal transfer from the central government to the municipality, the share of the rural population, conflict (abduction)[9], and municipal and year fixed effects. All fiscal variables data are expressed in real per capita Colombian pesos (COP) with the base year as 2008. The data period also overlaps with one of the most significant increases in oil prices, as shown in Figure 2.

Table 2 shows descriptive statistics for the variables used in regression analysis. There is a significant variation in population size, and that the share of the rural population is quite high at 60%[10]. Besides, only 6% of municipalities have oil extraction. All fiscal variables were measured as real per capita values in thousand 2008 Colombian pesos. During the 1993–2010 period, the average oil price was relatively high at about $55 per barrel, which was again driven by a sharp increase in oil prices.

**Table 2.** Descriptive Statistics.

| Variable | Mean | SD |
| --- | --- | --- |
| Total Expenditure | 507.92 | 441.15 |
| Tax revenue | 65.52 | 85.50 |
| Budget balance | −16.36 | 265.44 |
| Federal transfer | 390 | 268.46 |
| Population in thousand inhabitants | 39.08 | 214.278 |
| Share of Rural population | 58.22 | 23.61 |
| Conflict | 2.23 | 10.93 |
| Oil production status (Dummy-1990-1999) | 0.06 | 0.24 |
| Oil price (International) | 55.361 | 18.917 |
| Number of Observations | 12,023 | |
| Number of municipalities | 1093 | |

*Notes*: Descriptive statistics table provide averages over 2000–2010. All fiscal variables measure real per capita values in thousand 2008 COP.

## 4. Empirical Strategy

### 4.1. Fiscal Interaction Model

The Spatial Durban model (SDM) was used to investigate the spatial interaction of neighboring municipalities regarding three fiscal variables: budget balance, tax revenue, and public expenditure. The SDM has a vital standing in spatial econometrics. It has the advantage of including both the spatially lagged dependent and spatially lagged explanatory variables. In the SDM, a change in a particular explanatory variable in neighbor i directly affects that neighbor and has an indirect (spillover) effect on the remaining neighbors. Spatial regression specification in SDM tests the hypothesis as to whether or not spatial spillover leads to an erroneous conclusion LeSage and Pace (2009). The model to estimate the effect of neighboring municipalities' fiscal variables is given by

$$Y_{i,t} = \delta\,WY + \eta\left(Oil_i \times p_t^{oil}\right) + WX\theta + \alpha_i + \mu_{d,i} + \varepsilon_{d,i,t} \tag{2}$$

where $Y_{i,t}$ denotes the natural log of tax revenue and public expenditure and the budget balance level in municipality *i* at time *t*. WY is the fiscal interaction with a neighboring municipality, while W is an NXN spatial weight matrix. I used the 5-nearest neighbor (NN) criteria to construct a spatial weight matrix. According to this approach, the average number of municipalities has five neighbors. The estimation parameter $\delta$, which measures potential spatial interaction effects, is the interest coefficient in this paper. The dummy variable $oil_i$ was created by assigning one if oil extraction took place in the time frame of 1990 to 1999, before the period of analysis, otherwise it was assigned zero. The natural log of the world market price of oil $p^{oil}$ was converted to real Colombian pesos. The term X represents the control variables, the population of municipalities, the local share of the rural population, transfers from a higher level of government, and conflict (number of abductions). Also, $\varepsilon_{d,i,t}$ is the error term that is independent and identically distributed. $\alpha_i$

denotes municipalities' fixed effects, capturing time-invariant unobserved factors such as distance to the capital, climatic conditions, geographical size, culture, institutions, etc. The department-year fixed effects $\mu_{d,i}$ controls time variations that influence all municipalities within a department.

### 4.2. Commodity Price Shocks as Exogenous Variation in Fiscal Variables

The unobserved factors that change over time with variation within municipalities were included in both fiscal components (budget balance, spending, and tax revenue) in the neighboring municipalities and the error term $\varepsilon_{d,i,t}$. This covered the endogeneity problem in our regression model, which was addressed by using exogenous variation for fiscal variables with an instrument of a variable approach. Like (Acemoglu et al. 2013a), I used oil price shocks that affect the revenue for certain municipalities to different degrees due to the heterogeneous distribution of resources across Colombian municipalities. Colombia is a small exporter of oil in the international oil market, so municipalities in Colombia are price takers. Hence, the quasi-experimental identification strategy could be used in exploring exogenous variations in the global oil market.

In the 2SLS estimation, the first stage is given by the following equation

$$WY = \gamma\left(Oil_i \times p_t^{oil}\right) + \lambda\, W\left(Oil_i \times p_t^{oil}\right) + WX_k\Psi + X_{i,t}\ \rho + \alpha^f{}_i + \mu^f{}_{d,t} + \varepsilon^f{}_{i,t}. \tag{3}$$

$W(Oil_i \times p^{oil})$ is the interaction between neighboring oil endowment and the current international oil price in real Colombian pesos (COP), and was used as an instrument for neighboring budget balance, neighboring spending, and neigh boring tax revenue. For the validity of the instrument, two conditions needed to be satisfied. The first condition was for the instrument relevance, where the instrument $W(Oil_i \times p^{oil})$ must be correlated with the three-fiscal variables of the neighboring municipality (neighboring budget balance, spending, and tax revenue). I expected the instrument to be relevant for the three dependent variables because municipalities receive oil royalties from the federal government. Figure 1 shows the spatial variation in oil extraction between 1990 and 1999 before the period of analysis. We could see from Figure 2 that there is also a variation in the oil price over 2000–2010. The second condition for the validity of the instrument is the instrument exogeneity condition, which states that oil extraction in neighboring municipalities $W(Oil_i \times p^{oil)}$ must not be correlated with the error term $\varepsilon_{d,i,t}$, i.e., directly influence a municipality's fiscal outcomes. I controlled for the municipality fixed effects and time fixed effects. The local government policy extraction was not influenced because I used the oil extraction indicator from 1990 to 1999. I constructed the spatial weight matrix based on the 5-nearest neighbors (NN) because the average municipality has five neighbors for the spatial econometric analysis. In this weight matrix, every unit is assigned 5 nearest neighbors (5-NN). Also, equal weights are given to all neighbors, and the matrix is row normalized. Therefore, every non-zero elements of W are equal to 1/5.

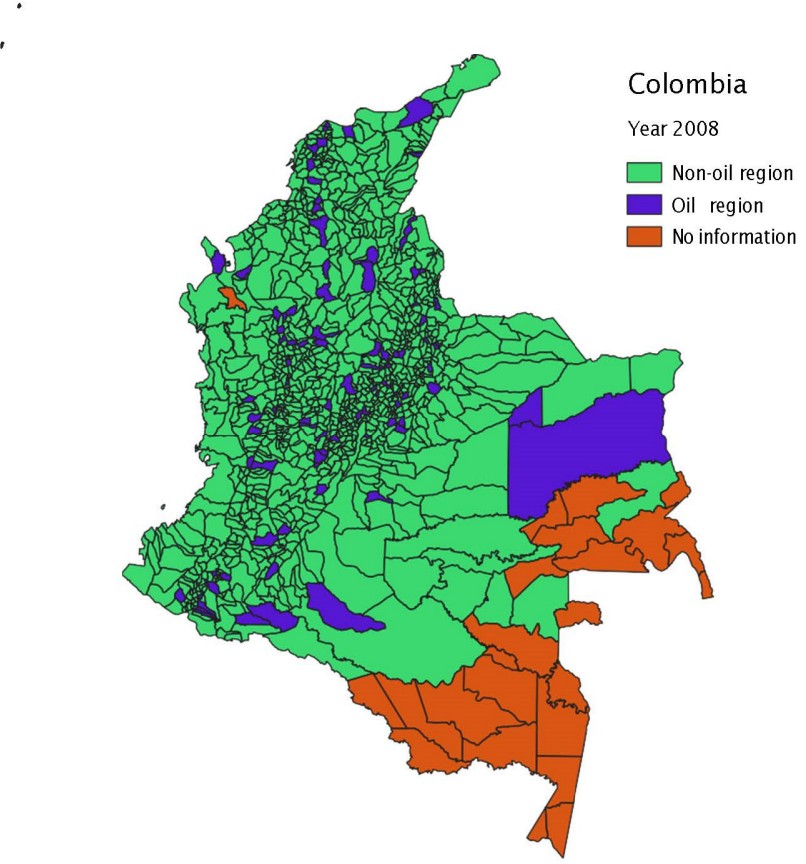

**Figure 1.** Oil producing vs. non-oil producing municipalities in Colombia. Source: Center for Studies on Economic Development (CEDE) at the University of the Andes.

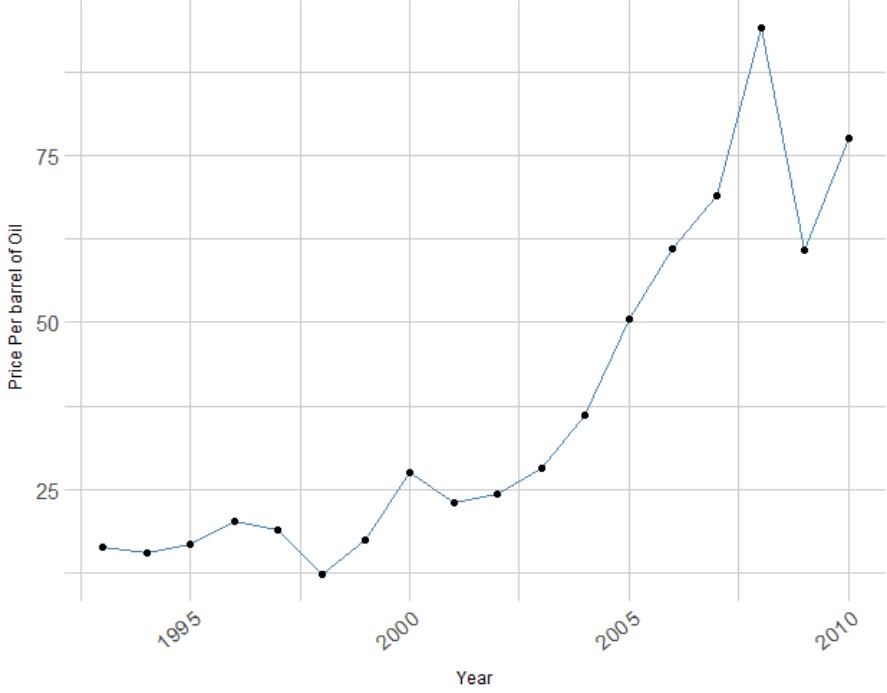

**Figure 2.** Oil prices in 2008 real COP (OPEC basket). Source: Center for Studies on Economic Development (CEDE) at the University of the Andes.

## 5. Empirical Results

### 5.1. Empirical Results Based on the Quasi-Experimental Instrument

Using the IV estimator, I obtained the second stage regression result of Equation (2) in Table 3; I present estimates for budget balance, total local public expenditure, and tax revenue in the second, third, and fourth columns. The first row of Table 3 represents the coefficient capturing spatial fiscal interaction effects. The results suggest no statistically significant interactions in budget balance and total public expenditure in the first two columns. But there is a significant interaction effect in tax revenue. Concerning tax competition, when the five-nearest neighboring municipalities increase their tax rate by 1%, it will cause the tax intake to increase by 1.02%. So, we see that there is only a spatial fiscal interaction effect in tax revenue. This suggests that while there is strong tax competition, municipalities do not mimic their neighbors when deciding whether to effect tax changes by changes in expenditures or debt. For the control variables, the results meet expectations, e.g., higher transfers from the central government increase the budget balance and total expenditure (Table 3). Population and population squared are not significant for the budget balance. However, they are substantial for both public spending and tax revenue. Also, the conflict variable[11] is statistically significant at a 5% level with positive and negative effects on three fiscal variables. The results indicate that municipalities have a higher budget balance and spend more when receiving royalties from oil extraction.

**Table 3.** Main IV estimation results for fiscal interaction from 2000–2010.

|  | Budget Balance | Public Spending | Tax Revenue |
|---|---|---|---|
| W_y | 0.261 | −0.198 | 1.018 * |
|  | (0.658) | (0.564) | (0.595) |
| Oil extraction x oil price | 0.733 * | 0.265 * | 0.191 |
|  | (0.432) | (0.081) | (0.134) |
| Population | 458.64 | 5.153 ** | −16.968 *** |
|  | (457.84) | (2.353) | (6.30) |
| Population squared | −0.081 | −2.528 * | 7.751 *** |
|  | (0.136) | (1.090) | (2.937) |
| Share of rural population | −169.59 | −0.549 *** | 0.110 |
|  | (209.12) | (0.138) | (0.279) |
| Federal transfer | 0.412 *** | 0.028 *** | 0.718 *** |
|  | (0.078) | (0.002) | (0.010) |
| Conflict | 0.577 *** | −3.206 *** | 0.005 *** |
|  | (0.277) | (0.801) | (0.001) |
| W_population | −279.76 | −0.772 | 13.08 |
|  | (864.10) | (5.559) | (12.94) |
| W_population squared | 0.001 | 0.648 | −6.03 |
|  | (0.081) | (2.569) | (5.994) |
| W_share of rural population | 0.105 | −0.477 | −0.239 |
|  | (312.32) | (0.435) | (0.613) |
| W_transfer | −0.156 | 0.012 | −0.723 * |
|  | (0.250) | (0.020) | (0.026) |
| W_cconflict | −0.016 | −3.741 | −0.002 |
|  | (0.819) | (2.47) | (0.004) |
| Observa1tions | 12,023 | 12,023 | 12,023 |
| Number of municipalities | 1093 | 1093 | 1093 |
| Municipality FE | YES | YES | YES |
| Department-year FE | YES | YES | YES |
| First stage F-statistics | 9.74 | 14.38 | 10.71 |

Robust standard errors in parentheses. *** $p < 0.01$, ** $p < 0.05$, * $p < 0.1$. Notes: This table shows the second stage 2SLS regression results for three dependent variables: public spending, tax revenue and budget balance. Total spending and tax revenue are transformed on log but budget balance is not.

It is also clear from Table 4, which shows the corresponding first stage result of the main estimation that the excluded instrument has sufficiently large F-statistics, i.e., the

excluded instrument is strong. The table shows that the coefficient of the interaction of neighboring municipalities' oil extraction with the oil price is positive and highly significant when neighboring budget balance, total spending, and tax revenue are the dependent variables in the first stage. This implies that municipalities' budget balance, tax revenue, and public spending are increased when they receive royalties from oil extraction.

**Table 4.** First stage regression results.

|  | **W_budget** | **W_Spending** | **W_tax** |
|---|---|---|---|
| W_Oil extraction x oil price | 0.932 ** | 0.248 *** | 0.472 *** |
|  | (0.298) | (0.066) | (0.144) |
| Oil extraction x oil price | −0.021 | −0.03 | −0.01 |
|  | (0.129) | (0.03) | (0.06) |
| Population | −138.95 | −0.49 | 0.081 |
|  | (226.16) | (1.268) | (2.27) |
| Population squared | 0.046 | 0.292 | −0.035 |
|  | (0.051) | (0.595) | (1.065) |
| Share of rural population | −17.81 | −0.01 | 0.066 |
|  | (63.66) | (0.055) | (0.127) |
| Federal transfer | −0.015 | 0.0001 | 0.001 |
|  | (0.009) | 0 | (0.002) |
| Conflict | 0.097 | 0.890 *** | 0.0005 |
|  | (0.134) | (0.272) | 0 |
| W_population | −336.98 | 6.761 *** | −9.766 ** |
|  | (438.07) | (2.254) | (4.46) |
| W_population squared | 0.083 ** | −3.141 *** | −4.692 ** |
|  | (0.03) | (1.05) | (2.089) |
| W_share of rural population | −156.85 | −0.623 *** | 0.221 |
|  | (148.89) | (0.118) | (0.278) |
| W_transfer | 0.387 *** | 0.035 *** | 0.729 *** |
|  | (0.004) | (0.002) | (0.005) |
| W_Conflict | 0.858 * | −3.314 *** | 0.005 |
|  | (0.347) | (0.819) | (0.001) |
| Observations | 12,023 | 12,023 | 12,023 |
| Number of municipalities | 1093 | 1093 | 1093 |
| Municipality FE | YES | YES | YES |
| Department -year FE | YES | YES | YES |
| F-statistics | 9.74 | 14.38 | 10.71 |

Robust standard errors in parentheses. *** $p < 0.01$, ** $p < 0.05$, * $p < 0.1$. Notes: This is the first stage regression result corresponding to Table 3.

### 5.2. Using Quasi Maximum Likelihood Approaches

For comparison to the IV estimates, the traditional quasi maximum likelihood estimator is used with municipality and time fixed effects, which does not use quasi-experimental variation for identification. I used the biased–correction QML estimator proposed by (Elhorts 2010). Table 5 provides the estimation results using the QML approach. From this estimator, there was still no spatial auto-correlation between neighboring municipalities in terms of the budget balance. However, it suggested a highly positive significant spatial autocorrelation in total local public expenditure, which was not significant when using the IV method before. It was clear from both the IV estimates and the QML estimates that there is a spatial interaction effect in tax revenue. The IV result for tax revenue is supported by the QML approach, although the magnitudes are very different. Also, for public expenditure, the estimated result using exogenous variation in the IV approach suggests that the spatial autocorrelation does not indicate a causal effect, while using the QML approach does indicate a causal effect.

**Table 5.** QML results for fiscal interaction.

| | Budget Balance | Public Spending | Tax Revenue |
|---|---|---|---|
| W_y | −0.002 | 0.084 *** | 0.052 *** |
| | (0.007) | (0.012) | (0.012) |
| W_o il extract ion x oil price | 0.025 | −0.109 | 0.33 |
| | (0.273) | (0.012) | (0.278) |
| Oil extraction x oil price | 0.850 * | 0.272 *** | 0.401 |
| | (0.449) | (0.080) | (0.131) |
| Population | 13.378 | 5.815 ** | −19.527 *** |
| | (307.67) | (2.538) | (5.711) |
| Population squared | 0.036 | −2.741 ** | 9.052 *** |
| | (0.021) | (1.163) | (2.69) |
| Share of rural population | 81.64 | −0.517 *** | 0.028 |
| | (184.47) | (0.143) | (0.252) |
| Federal transfer | 0.409 *** | 0.029 *** | 0.723 *** |
| | (0.074) | (0.002) | (0.009) |
| Conflict | 0.543 ** | −0.001 * | 0.004 *** |
| | (0.220) | (0.001) | (0.131) |
| W_popula tion | 1458.64 | −3.746 | −8.652 |
| | (1929.19) | (4.18) | (9.080) |
| W_population squared | −0.098 | 1.907 | 3.946 |
| | (0.122) | (1.94) | (4.229) |
| W_shar e of rural population | 689.55 ** | 0.188 | 0.063 |
| | (312.32) | (0.193) | (0.461) |
| W_transfer | 0.025 | −0.004 | −0.035 *** |
| | (0.032) | (0.003) | (0.012) |
| W_conflict | 0.216 | 0.001 * | −0.000 |
| | (0.868) | (0.009) | (0.003) |
| Observations | 12,023 | 12,023 | 12,023 |
| Number of municipalities | 1093 | 1093 | 1093 |
| Municipality FE | YES | YES | YES |
| Department -year FE | YES | YES | YES |
| First stage F-statist ics | 9.74 | 14.38 | 10.71 |

Robust standard errors in parentheses. *** $p < 0.01$, ** $p < 0.05$, * $p < 0.1$.

### 5.3. Robustness Checks

In applied econometric models to reduce heteroscedasticity or minimize the effect of outliers, transformed variables are often used. One way of transforming variables is taking the logarithm of variables. The log transformation is applicable only when the dependent variable is strictly positive. I did not use log transformation for budget balance in Equations (2) and (3) because of negative observations.

For a robustness check of the result presented above, I used the inverse hyperbolic Sine transformation (IHS)[12] method for the budget balance (Zhang et al. 2000). The IHS transformation is applicable in a regression analysis where the dependent variable to be transformed may be positive, zero, or negative. The conclusion I found by using IHS was consistent with the previous finding. So, from Table 6, it is clear that there is no spatial interaction effect on the budget balance between the municipalities. A caveat of the IV estimate in this robustness check is that the instrument becomes weak.

**Table 6.** Robustness checks: using inverse hyperbolic sine (IHS) transformation.

| | Budget Balance (IV) | Budget Balance (QML) |
|---|---|---|
| W_y | 5.529 | 0.003 |
| | (7.129) | (0.0011) |
| W_oil extraction X oil price | −7.327 | −0.664 |
| | (10.634) | (1.787) |
| Oil extraction x oil price | 0.371 | 0.272 *** |
| | (0.449) | (0.08) |
| Population | 54.743 | 253.74 ** |
| | (420.81) | (121.65) |
| Population squared | −23.69 | −126.78 ** |
| | (210.49) | (60.61) |
| Share of rural population | −1.02 | −2.869 |
| | (6.17) | (2.206) |
| Federal transfer | 0.247 | 0.489 *** |
| | (0.367) | (0.037) |
| Conflict | −0.082 | 0.025 |
| | (0.167) | (0.034) |
| W_population | −2848.86 | 401.03 * |
| | (4051.31) | (225.82) |
| W_population squared | 1418.998 | −198.95 ** |
| | (2019.67) | (111.58) |
| W_share of rural population | 37.18 | 2.69 |
| | (53.69) | (4.14) |
| W_transfer | −3.035 | 0.216 ** |
| | (4.083) | (0.096) |
| W_conflict | −0.379 | 0.001 |
| | (0.682) | (0.062) |
| Observations | 12,023 | 12,023 |
| Number of municipalities | 1093 | 1093 |
| Municipality FE | YES | YES |
| Department -year FE | YES | YES |
| First stage F-statistics | 2.37 | |

Robust standard errors in parentheses. *** $p < 0.01$, ** $p < 0.05$, * $p < 0.1$.

## 6. Conclusions

There is much literature on jurisdiction competition in North America and Western European countries; little is known on its presence and effect in developing countries. This paper contributes to filling this gap in the literature by examining whether jurisdictional competition has been present in Colombia after major decentralization. This study investigated fiscal policy variables' spatial interaction—budget balance, public expenditure, and tax revenue—using local level panel data of 1093 Colombian municipalities. A quasi-experimental identification strategy was used to identify the spatial interaction effects. The estimation depended on the municipalities' exogenous shock in oil prices on the world market due to the municipalities' local oil endowments.

The quasi-experimental results showed a significant positive spatial interaction in local taxes, but there were no significant spatial interactions in the local budget balance and local public expenditures. When the spatial fiscal interactions was estimated using traditional methods without using a quasi-experimental variation, it suggested significant spatial auto-correlation not only in local taxes but also in local public expenditures. This difference in results showed that the quasi-experimental variation is necessary to identify causal fiscal interaction effects. The causal results suggest that Colombian municipalities engage in tax competition. However, they do not mimic their neighbors when it comes to spending and borrowing. This implies that Colombia's decentralization has not led to a race to the bottom concerning local public expenditures or mutually accelerated borrowing. These results encourage decentralization reform in developing countries as far as tax competition is not considered to be harmful.

For policy implications, this paper provides the relevance of the fiscal competition model in developing countries after decentralization reforms. Also, this paper's contribution will have policy implications for South American countries that have adopted a similar fiscal policy regarding royalties from natural resources and their distribution to subnational governments. It is challenging to find a valid instrument to perform a quasi-experimental identification strategy with exogenous variation. So, more studies are needed to check whether this finding is consistent with others developing world where natural resources are not available as a form of income for local governments.

**Funding:** This research received no external funding.

**Conflicts of Interest:** The author declares no conflict of interest.

## Notes

1   According to Faguet and Sanchez (2014) the decentralization has made Colombian mayors more accountable.
2   According to (Meyer 1995) quasi-experimental econometrics is used to mimic random assignment either by controlling for the variation of the treatment variable or by controlling for the assignment mechanism itself. Instrumental variable (IV), Difference-in-Difference (DID), the propensity score matching (PSM), and Regression discontinuity (RD) are frequently used quasi-experimental econometrics tools.
3   Municipalities endowments with oil resources were measured during 1990–1999.
4   In this study, 1093 municipalities were considered out of 1122, 20 units are dropped when they maintain a special territorial status different from that of municipalities situated in remote and sparsely populated areas, four municipalities were dropped which were established after 2007, and five municipalities who do not have direct neighbors were dropped.
5   There are no systematic trends of missing values in the data set. I used linear interpolation to impute missing values that are recorded less with than 3% of the observations.
6   Law 358, 1997.
7   Law 617, 2001.
8   If liquidity is less than 40% and solvency is less than 80%, those municipalities will have a "green light" status, and are allowed to borrow without federal intervention. If liquidity is between 40% and 60% and solvency is less than 80%, those municipalities will have a "yellow light" status, meaning they can negotiate debt contracts but need authorization from the Ministry of Finance. If liquidity is greater than 60% and solvency is greater than 80%, those municipalities will have a "red light" status, and are not allowed to borrow.
9   It is often said that 60% of all kidnappings in the world take place in Colombia. About half of the abductions are attributed to leftist guerrillas, and this proportion may be much higher.
10   Colombia is a primarily urban country. According to 2005, 45% of the country's population lives in the 20 largest cities.
11   The Colombian civil war started in the 1960s and was severe during 1990–2000. Internal conflicts can influence spending patterns, tax revenue, and budget balance at a local level.
12   Inverse Hyperbolic Sine Transformation (IHS): For any random variable x, the IHS is Defined by $\operatorname{archsinh}(x) = \ln(x + \operatorname{sqrt}(x^2 + 1))$.

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
