# Peer review of "Spatial Fiscal Interactions in Colombian Municipalities: Evidence from Oil Price Shocks"

_jrfm, doi:10.3390/jrfm14060248_

Round 1

Reviewer 1 Report

The paper should extend the research after 2010 and if this is not possible, explain why this is the case and which is the relevance of choosing only that time frame. Is this time frame relevant? 

Other minor changes I recommend are to change subjective expressions like me, my with more objectives ones referring to the paper, the research, the study. The tense of the verbs used should be mostly present tense not past tense. The former is mostly used in research papers. 

Source 31 does not have a year

I suggest adding newer references from 2019-2021. 

Reviewer 2 Report

The topic of the paper is interesting and the empirical analysis presented through the paper is well justified. The most notable contribution of the paper consists in investigating the spatial fiscal interactions in Colombian municipalities, using quasi-experimental identification strategy exploiting exogenous variation from global oil price shocks. Indicators like budget balance, tax revenue and public spending were used, based on a panel data approach.

Even though the empirical study has potential and the methodology is relevant and well applied, the paper requires major revisions:

  1. The Introduction section needs a better statement of the contribution of the paper among the international literature review.

  1. Both the Empirical results section and the Concluding section need to be extended to better describe the main findings and their contribution to the international literature review in the field. Moreover, the author should argue how the paper manages to fill in the gap in the field and the policy implications of the research results.

Round 2

Reviewer 2 Report

The author has improved the paper by addressing all my previous comments. Thus, in my opinion, the paper can now be published in its current form.